# Portable, Rapid, and Sensitive Time-Resolved Fluorescence Immunochromatography for On-Site Detection of Dexamethasone in Milk and Pork

**DOI:** 10.3390/foods10061339

**Published:** 2021-06-10

**Authors:** Xiangmei Li, Xiaomin Chen, Jinxiao Wu, Zhiwei Liu, Jin Wang, Cuiping Song, Sijun Zhao, Hongtao Lei, Yuanming Sun

**Affiliations:** 1Guangdong Provincial Key Laboratory of Food Quality and Safety, College of Food Science, South China Agricultural University, Guangzhou 510642, China; lixiangmei12@163.com (X.L.); chenxiaomin2021@163.com (X.C.); liuzhiwei2021888@163.com (Z.L.); wj646607936@163.com (J.W.); 2Shanxi Institute of Feed and Veterinary Drug control, No. 5 Shengli West Street, Jiancaoping District, Taiyuan 030000, China; wdwk030@163.com; 3China Animal Health and Epidemiology Center, 369 Nanjing Rd, Si Fang Qu, Qingdao 266032, China; wdwk045@163.com (C.S.); lixiangmei12@scau.edu.cn (S.Z.)

**Keywords:** dexamethasone, time-resolved fluorescent microspheres, immunochromatographic assay, milk, pork

## Abstract

Dexamethasone (DEX) is widely used because of its anti-inflammatory, anti-endotoxin, anti-shock, and stress-enhancing response activities. It can increase the risk of diabetes and hypertension if it is abused or used improperly. However, there is a lack of sensitive and rapid screening methods for DEX in food. In this study, a time-resolved fluorescent microspheres immunochromatographic assay (TRFM-ICA) integrated with a portable fluorescence reader was developed for the quantitative detection of DEX in milk and pork. The cut-off values of the TRFM-ICA were 0.25 ng/mL and 0.7 µg/kg, respectively. The limits of quantitation (LOQs) were 0.003 ng/mL and 0.062 µg/kg, respectively. The recovery rates were 80.0–106.7%, and 78.6–83.6%, respectively, with the coefficients of variation ranging 6.3–12.5%, and 7.5–10.3%, respectively. A parallel experiment for 20 milk and 10 pork samples with LC-MS/MS was carried out to confirm the performance of the on-site application of the developed TRFM-ICA. The results of the two methods are basically the same; the correlation (R^2^) was >0.98. The establishment of TRFM-ICA will provide a new sensitive and efficient technical support for the rapid screening of DEX in food.

## 1. Introduction

Dexamethasone (DEX) is a synthetic corticosteroid and has pharmacological effects including anti-inflammatory, anti-toxic, anti-allergic, and anti-rheumatic activities [1]. Therefore, it is widely used in veterinary clinical treatment of maternal metabolic diseases or in combination with antibiotics to treat infectious diseases, and it is also one of the commonly used drugs in livestock and poultry breeding [2]. However, DEX can also cause certain adverse reactions to animals, such as gastrointestinal reactions, allergic reactions, liver dysfunction, skin and mucosal symptoms, etc. Long-term consumption of animal products with excessive DEX will cause diabetes, hypertension, myocardial infarction, gastrointestinal ulcer, and other symptoms [3]. Therefore, DEX is strictly forbidden to be used as a growth hormone in animal-derived food globally [4]. Many countries and organizations have established the maximum residue limits (MRLs) for DEX in animal foods. For example, China and Codex Alimentarius Commission set MRLs of 0.3 and 0.75 µg/kg in milk and pork, respectively [5,6]. Based on the wide application, serious side effects, and trace detection requirements of DEX, it is imperative to establish a rapid and sensitive detection method for DEX in food to ensure the health of humans and animals.

Since the emergence of immunochromatographic assay (ICA), it has become the most popular rapid detection method for food safety testing due to its outstanding advantages such as simple sample preparation, fast acquisition of test results, no professional training, low cost, and being suitable for screening large quantities of samples [7,8,9]. ICA based on gold nanoparticle (GNP) is the most widely used method on the market. However, with the increasing demand for food safety detection, the sensitivity of GNP-ICA has become a bottleneck restricting its development [10]. Therefore, how to improve the sensitivity of ICA has become the focus of research. There are mainly two ways to overcome this deficiency from the published literature. On the one hand, novel Ab-labeled tracers are synthesized for signal amplification, such as quantum dots (QD) [11,12], fluorescent microspheres (FM) [8,13], chemiluminescent materials [14,15], up conversion phosphorescence (UCP) [16], biotin-affinity [17], and metal–organic frameworks [18]. On the other hand, external analytical instruments are developed, such as desktop [19,20], hand-held [21,22], or smartphone-based [23,24] reading platforms. Therefore, with the continuous development of material technology and equipment, the problem of ICG restricted by sensitivity will be alleviated, and its advantages in rapid detection will become increasingly prominent. However, up to now, there have been only three reports on the detection of DEX in animal-derived foods by ICA. One used GNP-ICA for milk detection, where the cut off value was 0.5 µg/kg, which could not meet the requirement of residue detection [25]. The second used UCP-ICA for the animal tissue detection, and the stability of UCNP is controversial [26]. The third use latex microspheres (LMs)-ICA for milk and pork detection; based on the color diversification of LMs, different samples could be distinguished by color. However, the sensitivity was not as good as our present work [27].

Therefore, in order to provide a stable, sensitive, reliable, and rapid detection method for DEX residue detection, ICG based on time-resolved fluorescent microspheres (TRFM) was established. TRFM was employed as the signal-labeled tracer with several advantageous features: (a) Rare earth ions with longer fluorescence half-life (103–106 times the traditional fluorescence) are used as labels, which have extremely wide stokes shift (he excitation wavelength and emission wavelength are 365 and 610 nm, respectively, and the stoke shift is more than 200 nm) and long fluorescence quenching time, thus effectively eliminating the interference of various non-specific fluorescence and improving the accuracy and sensitivity [24]. (b) There are thousands of fluorescent molecules in the TRFM, which greatly improves the labeling efficiency of fluorescence and analytical sensitivity [28]. (c) The surface of TRFM is modified with carboxyl or other functional groups, which are used for covalent coupling with proteins or Ab, improving the stability of the conjugates. These features are ideal for the development of ICG. Meanwhile, a portable, compact desk reader was used to quantify results. This integrated strategy could provide valuable technical support for the on-site detection of DEX in animal-derived food.

## 2. Materials and Methods

### 2.1. Reagents and Instruments

DEX, hydrocortisone, prednisone, triamcinolone, betamethasone, 1-ethyl-3-(3-dimethylaminopropyl) carbodiimide hydrochloride (EDC), N-hydroxysuccinimide (NHS), 2-(N-morpholino) ethanesulfonic acid (MES), ovalbumin (OVA), and bovine serum albumin (BSA) were purchased from Sigma-Aldrich (St. Louis, MO, USA). TRFM, europium chelates (365/610), with 1% solid content (*w/v*) and 0.2 µm particle size, was purchased from Bangs Laboratories, Inc (Fishers, IN, USA). Anti-DEX monoclonal A and DEX-OVA coating antigen (Ag) were prepared in our laboratory. The nitrocellulose filter (NC) membrane (Sartorius, UniSart CN95) was purchased from Sartorius Stedim Biotech GmbH (Goettingen, Germany). The microtiter plates were supplied by the JET BIOFIL Co. (Guangzhou, China). The polyvinylchloride (PVC) backing plate (SMA31-40), sample pad (GF-2), and absorbent pad (CH37) were purchased from Shanghai Kinbio Tech. Co., Ltd. (Shanghai, China). Sucrose, sodium chloride, and other chemicals reagents were bought from Sinopharm Chemical Reagent Co., Ltd. (Shanghai, China).

The strip cutter ZQ-2000 and the slitting machine SPT300 were purchased from Shanghai kinbio Tech. Co., Ltd. (Shanghai, China). The XYZ-3060 Dispensing Platform was bought from BioDot, Inc. (Irvine, CA, USA). The UV spectrometer was provided by Thermo Fisher Scientific Co. (Waltham, Massachusetts, USA). The Lynx-4000 centrifuge was obtained from Thermo Fisher Scientific GmbH (Berlin, Germany). The time-resolved fluorescence quantitative analysis reader (FQ-S2, 254 nm, 365 nm) was purchased from WDWK Biotechnology Co., Ltd. (Beijing, China).

### 2.2. Preparation of TRFM Immunoprobe

The preparation of TRFM immunoprobe mainly includes two steps: activation of carboxyl groups on the surface of TRFM and covalent coupling with DEX Ab [28] (Figure 1A). Briefly, with constant stirring (300 rpm/min), 0.1 mg of TRFM was added in 1 mL of MES buffer (50 mM, pH 5.5). Then, 15 µL of freshly prepared 0.5 mg/mL EDC and NHS solution were sequentially added. The mixture was centrifuged at 14,000× *g* for 15 min at 4 °C after reaction for 15 min. The supernatant was discarded, and the bottom sediment was redissolved with 1 mL of borate buffer (BB, 50 mM, pH 8.0). Anti-DEX Ab (1 µL, 1.0 mg/mL), which was dissolved in 60 µL of BB (2 mM, pH 8.0), was added. The reaction solution was well mixed and incubated at room temperature for 45 min, and then 20 µL of 20% BSA (*w/v*) were added for blocking. The above solution was centrifuged at 14,000× *g* for 15 min at 4 °C after another 60 min of blocking reaction. The bottom sediment was dissolved in 200 µL of resuspension, which was stored at 4 °C for later use. The key technical parameters are shown in Appendix A.

### 2.3. Preparation of the Test Strips

The goat anti-mouse secondary Ab and coating Ag (DEX-OVA) were diluted to the optimal concentration, and then sprayed on the NC film to form the control (C) and test (T) lines, respectively. The technical parameters of the spray film were as follows: spray length, 30 cm; distance between T-C line, 8 mm; and spray volume, 0.8 µL/cm. The processed NC films were placed in a 37 °C oven to dry overnight. The sample pads were immersed in the designed sample pad treatment solution for 30 s, and then dried for 2 h at 60 °C. Finally, the dried NC film, sample pad, and absorbent pad were pasted on the PVC backing pad with 2 mm overlap each other (Figure 1A). The PVC sheet was cut into 3.5 mm strips for use. The working principle is that the analyte competes with coating Ag for the limited Abs. The binding of the analyte to the Abs inhibited that of the coating Ag to the Abs, which was judged by the fluorescence intensity of T-line. The greater is the amount analyte, the weaker is the T-line signal, or there is even no color (Figure 1B). The working parameters of the goat anti-mouse secondary Ab and coating Ag are shown in Appendix A.

### 2.4. Sample Pretreatment

Milk: Milk samples can be detected directly without pretreatment. If the sample is rich in fat, the fat can be removed by centrifugation.

Pork: After the pork was chopped with scissors and homogenized, 4 g were accurately weighed into a centrifuge tube, and 4 mL of 0.2 M phosphate buffer (PB, pH 7.4)-methanol solution (*v/v*) were added. The sample was vortexed vigorously for 3 min, sonicated for 3 min, and then centrifuged at 4000× *g* for 10 min at room temperature. The supernatant was transferred out for use.

### 2.5. Test Procedure

Five microliters of TRFM-DEX Ab immunoprobe were added to the microwell, and then 150 µL of standard solution or sample solution were added. After reaction for 4 min, the test strip was inserted into the microwell for 5 min for chromatographic reaction. The test strip was taken out of the microwell, and the sample pad was quickly peeled off. The qualitative result was observed under an ultraviolet lamp (Figure 1C). The quantitative results were obtained by inserting the test strip into the fluorescence quantitative analysis reader (Figure 1C). The fluorescence intensity of the T/C lines were converted into the corresponding peak. The stronger is the signal, the larger is the peak area.

### 2.6. Method Performance Evaluation

#### 2.6.1. Sensitivity

In this study, the spiked milk and pork samples with a series of DEX concentrations were used to assess the sensitivity of the developed TRFM-ICG; each spiked concentration was detected in triplicate. The sensitivity was expressed by the cut-off value, calibration curve, and limit of quantitation (LOQ). The cut-off value was defined as the lowest DEX concentration that can cause the T-line signal to disappear. The fluorescence intensity of the T/C line can be achieved by the fluorescence reader. The DEX concentration in milk or pork sample was quantified by the calibration curve, taking the concentration of DEX as the x-axis and B/B_0_ (T/C line fluorescence signal ratio, where B is with analyte in the reaction system and B_0_ is without analyte in the reaction system) as the y-axis. The LOQ was defined as the concentration corresponding to 80% of B/B_0_ value on the calibration curve [29].

#### 2.6.2. Specificity

Method specificity is expressed by cross-reactivity (CR). DEX and several structural analogs, such as hydrocortisone, prednisone, triamcinolone, and betamethasone, were analyzed at different concentrations for the CR by the indirect competitive enzyme-linked immunosorbent assay (icELISA) and TRFM-ICG. The CR is calculated by the ratio of the IC_50_ of the target analyte/analogs.

#### 2.6.3. Accuracy and Precision

Milk and pork samples, identified by LC-MS/MS as DEX-free, were spiked with three known concentrations of DEX, respectively. All spiked samples were processed according to the previous description. Each spiked level was tested in triplicate on three different days using the fluorescence reader. The precision and accuracy of the TRFM-ICG were assessed by the coefficient of variation (CV) and recovery, respectively.

### 2.7. Blind Sample Detection

Twenty milk and ten pork blind samples were provided by Guangdong Provincial Key Laboratory of Food Quality and Safety, which were tested by the developed TRFM-ICG and LC-MS/MS as the confirmatory method. It should be noted that we did not know the concentration of DEX in each sample. Each sample was tested in triplicate; the correlation between the two methods was compared; and the consistency of the results reflects the reliability of the method.

The working parameters of the LC-MS/MS method are given in the Appendix A. The sample pretreatment of milk and pork was consistent with the national standards [30,31].

## 3. Results and Discussion

### 3.1. Optimization of the TRFM-ICG

Several important technical parameters were optimized to achieve the best detection performance of the established TRFM-ICG, including the particle size and activated pH value of TRFM, the optimal pH value and ion concentration of Ab coupling with TRFM, Ab amount, sample pad treatment solution formula, etc.

#### 3.1.1. Particle Size of TRFM

The particle size of the microsphere determines the specific surface area, which affects not only the binding efficiency of the carboxyl groups on the surface of the microsphere with Ab but also the chromatographic release of the immunoprobe [8,28]. The detection results of 200 and 300 nm microspheres were compared. The results show that the release effect of the 300 nm microsphere immunoprobe was not as good as that of the 200 nm microsphere immunoprobe. There were many residues in the reaction zone, which led to the background being very red and fuzzy (Appendix A). However, high background values will have a negative impact on the subsequent visual judgment and quantitative detection. Therefore, 200 nm microspheres were selected as DEX-Ab labeling tracers.

#### 3.1.2. Activation pH Value of TRFM

TRFM are modified with carboxyl groups and need to be activated before coupling with Ab. The appropriate pH value can improve the activation efficiency of carboxyl group, and thus improve the binding efficiency with Ab [24]. The carbodiimide method was applied to activate the carboxyl groups under four different pH conditions, including pH 5.0, 5.5, 6.0, and 6.5. The results show that the fluorescence signal intensity of T-line increased with the increase of pH value, but the inhibitory effect was difficult to distinguish with the naked eyes. With the help of quantitative analysis results, the inhibition rates were the highest in the detection of milk and pork samples under pH 5.5 (Appendix A). Therefore, the optimal activation pH was selected as 5.5.

#### 3.1.3. The Ab Dilution Buffer

The Ab dilution buffer can not only maintain the biological activity of Ab but also prevent non-specific reactions and improve the sensitivity of the method [7,32]. Five different Ab dilution buffers, namely ultrapure water, 0.01 M PB (pH 7.4), 0.01 M PB (pH 7.4, 0.5% BSA), 0.5% BSA, and 0.002 M BB (pH 8.0), were used to dilute DEX Ab, adding a control group with no Ab diluent. The photos of the test strips and quantitative detection results show that 0.002 M BB (pH 8.0) as an Ab diluent had outstanding advantages in inhibition effect, and the fluorescence intensity was also satisfactory (Figure 2). This result shows that pH was a key factor affecting the binding of Ab-FMs, and BSA would hinder the coupling of Ab-FMs. Therefore, BSA should not be contained in the Ab diluent. Interestingly, this result is exactly the opposite of our previous research [27]. Therefore, there was no doubt that 0.002 M BB (pH 8.0) was our target Ab dilution.

#### 3.1.4. The Ab Amount

The Ab amount plays a decisive role in the signal intensity and inhibitory effect of the immunoassay method [10,33]. The detection performance of four different Ab amounts (0.8, 1.0, 1.2, and 1.4 µg for milk and 0.6, 0.8, 1.0, and 1.2 µg for pork) were investigated. As shown in Appendix A, with the increased of Ab amount, the fluorescent signal of T-line increased gradually, but the inhibition effect became worse. Combined with the results of quantitative detection, it was not difficult to find that, when the amount of Ab was 0.8 µg, which was equivalent to adding 80 µL of Ab solution, the inhibition rate of the test strip was the best. Therefore, the optimal amount of Ab was 0.8 µg for both milk and pork detection.

#### 3.1.5. Key Reagents of Sample Pad Treatment Solution

The main function of the sample pad is the carrier of the sample, which can promote the release of Ab probe and eliminate the matrix interference of the sample [24,28]. Therefore, the handling of the sample pad is very important.

Surfactant. Surfactant can promote the release of immunoprobe, affect the behavior of immunoprobe on chromatographic pads, and reduce non-specific reaction [28,34]. In this study, we employed Tween-20 as a surfactant and studied the effect of its concentration changes on the performance of test strips. As shown in Appendix A, we prepared a series of different concentrations of Tween-20 in the sample pad treatment solution and found that, as the concentration of Tween-20 increased, the release rate of immunoprobe accelerated. The results of quantitative analysis show that the inhibition rate was the best when the concentration of Tween-20 was 0.5% for milk samples and 0.2% for pork samples. The reason for the inconsistency of the optimal concentration should be closely related to the viscosity and fluidity of the samples.

Sample pad treatment buffer. It is particularly important for the sample pad treatment buffer to protect Ab activity and eliminate the interference of the sample matrix [24,29]. In this study, the effects of three kinds of buffers with different ionic strength and pH values on the detection performance of test strips were compared. The results (Figure 3) show that pH value > 9.0 was not conducive to the performance of Ab activity because the fluorescence intensity of test strip was not ideal. However, high ionic strength was conducive to inhibition. Considering the fluorescence signal intensity, inhibition effect, and inhibition rate, 0.05 M PB was finally determined as the buffer of the sample pad treatment solution.

To sum up, the formula of the sample pad treatment solution for milk detection was 0.05 M PB (pH 7.4, 0.5% Tween-20, 0.3% PVP) and for pork detection was 0.05 M PB (pH 7.4, 0.2% Tween-20, 0.3% PVP).

### 3.2. Method Performance Evaluation

#### 3.2.1. Sensitivity

Based on the above optimization conditions, negative milk and pork samples were spiked with DEX at different concentrations and tested by the optimized TRFM-ICG. As shown in Figure 4, the cut-off values of TRFM-ICG for DEX in milk and pork were 0.25 ng/mL and 0.7 µg/kg, respectively. The LOQ were 0.003 ng/mL and 0.062 µg/kg, respectively. The dynamics ranged 0.05–0.3 ng/mL and 0.12–1.35 µg/kg, respectively.

#### 3.2.2. Specificity

The specific results are shown in Appendix A. DEX Ab exhibited a strong CR to triamcinolone, with a CR rate of 54.5%, which may be due to the similar structure of the Ag recognition site. The CR rates to betamethasone, prednisolone, and hydrocortisone were 24.0%, 14.0%, and 1.5%, respectively. The specificity results of TRFM-ICG were consistent with that of icELISA, and the CR rates to DEX, triamcinolone, betamethasone, prednisolone, and hydrocortisone were 100%, 62.5%, 31.3%, 20.8%, and 4.2%, respectively. The specificity results show that the established TRFM-ICG method can be used for the multiple targets’ detection of five glucocorticoids.

#### 3.2.3. Accuracy and Precision

The DEX-free milk and pork samples were spiked with DEX standard working solution so that the concentrations of DEX in milk were 0.075, 0.15, and 0.3 ng/mL and in pork were 0.35, 0.7, and 1.4 µg/kg. The recovery rates of DEX in milk and pork samples were 80.0–106.7%, and 78.6–83.6%, respectively, with the CVs of 6.3–12.5%, and 7.5–10.3%, respectively (Table 1). The recoveries and CVs meet the requirement for residue detection. These results indicate that the established TRFM-ICG method has good accuracy and reproducibility.

### 3.3. Blind Sample Detection

Twenty milk and ten pork blind samples were analyzed simultaneously with our established TRFM-ICG method and the national standard method (LC-MS/MS). The test results are shown in Table 2 and Figure 5. Eighteen milk and eight pork samples were detected to contain DEX using LC-MS/MS, and the same results were obtained by TRFM-ICG. The detection results of the two methods were basically consistent; the correlation coefficient was greater than 0.98 (R^2^ > 0.98). These results indicate that the established TRFM-ICG method was accurate and reliable, and it can be used in the actual detection.

### 3.4. Comparison of DEX Immunoassay

The European Union, Japan, China, and many other countries and organizations have clearly stipulated the MRLs of DEX in animal-derived food [5,6,35,36]. Unfortunately, the detection methods of DEX residue in animal-derived food are rare, and there are even fewer rapid immunoassay methods. Until now, there are only six reports on immunoassay methods for DEX in foods (Table 3), of which three are ELISA methods [37,38,39] and the other three are ICG methods [25,26,27]. As is known, compared to ICG, the operation process and sample pretreatment of the ELISA method are relatively cumbersome. The outstanding advantage of ICG is that it is simple and fast, and the results can be achieved within 5–10 min. Therefore, the development of ICG can greatly improve the screening efficiency of DEX, and it is a useful supplement to monitoring methods.

Among the three reported ICG methods, one used traditional colloidal gold as the Ab tracer, the detection sample was milk, and the sensitivity could not meet the requirements for the detection of DEX residues [25]. The second used UCP as the Ab tracer, only qualitative detection was carried out on animal tissue sample, and the stability of UCP is controversial [26]. The third was the work of our team. The advantage of that work was that we could use LMs with different colors to distinguish different samples. However, the sensitivity of that method was not as good as the one in our current work [27]. This may be because there are thousands of fluorescent molecules in TRFM, which greatly improves the labeling efficiency of fluorescence and analytical sensitivity.

## 4. Conclusions

In this study, a rapid, sensitive TRFM-ICG method based on a portable fluorescence reader was firstly established and confirmed for screening detection of DEX in milk and pork. TRFM was employed as the Ab tracer, and the cut-off values for DEX in milk and pork were 0.25 ng/mL and 0.7 µg/kg, respectively. The LOQs were 0.003 ng/mL and 0.062 µg/kg, respectively. The recovery rates ranged 80.0–106.7%, and 78.6–83.6%, respectively, with the CVs ranging 6.3–12.5% and 7.5–10.3%, respectively. The results could be obtained within 10 min. Parallel testing of blind milk and pork samples with LC-MS/MS demonstrated that the developed quantitative TRFM-ICG method was accurate, reliable, and user-friendly. The establishment of the TRFM-ICG method can provide a new, efficient, and useful technical support for the rapid screening of DEX in food.

## Figures and Tables

**Figure 1 foods-10-01339-f001:**
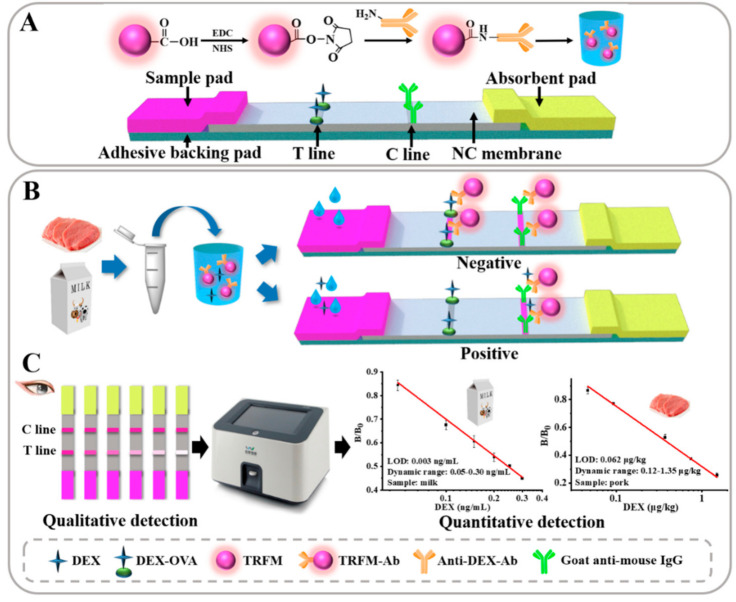
Schematic diagram of TRFM-ICG for DEX detection in milk and pork: (**A**) the preparation principle of TRFM-DEX Ab immunoprobe and structure of test strip; (**B**) the detection principle of test strip; and (**C**) the qualitative and quantitative test results.

**Figure 2 foods-10-01339-f002:**
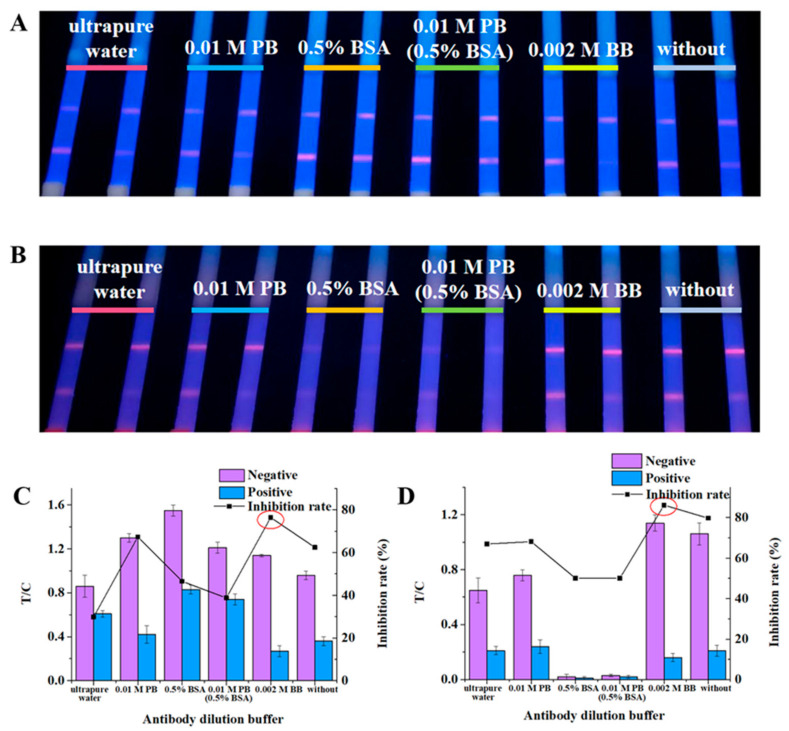
The fluorescence intensity, inhibition effect, and inhibition rate results of the Ab dilution buffer: (**A**) ultraviolet lamp results for milk detection; (**B**) ultraviolet lamp results for pork detection; (**C**) fluorescence quantitative results for milk detection; and (**D**) fluorescence quantitative results for pork detection.

**Figure 3 foods-10-01339-f003:**
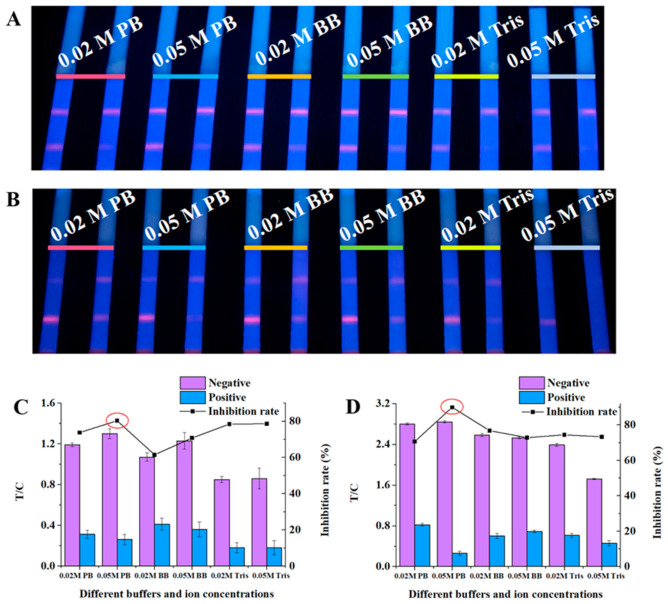
The fluorescence intensity, inhibition effect, and inhibition rate results of the sample pad treatment buffer: (**A**) ultraviolet lamp results for milk detection; (**B**) ultraviolet lamp results for pork detection; (**C**) fluorescence quantitative results for milk detection; and (**D**) fluorescence quantitative results for pork detection.

**Figure 4 foods-10-01339-f004:**
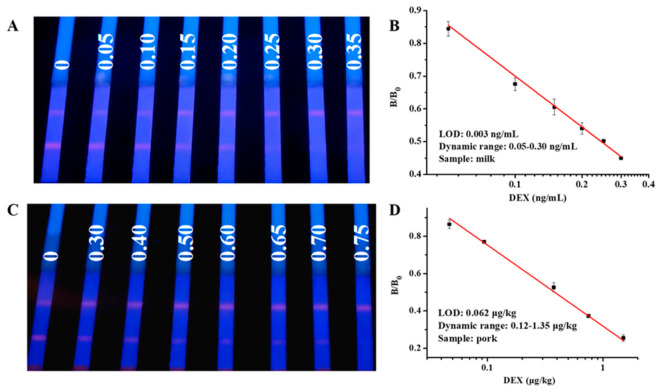
The results of TRFM-ICG for milk and pork detection: (**A**) the cut-off value for milk was 0.25 ng/mL; (**B**) calibration curve for quantitative detection of DEX in milk; (**C**) the cut-off value for pork was 0.7 µg/kg; and (**D**) calibration curve for quantitative detection of DEX in pork.

**Figure 5 foods-10-01339-f005:**
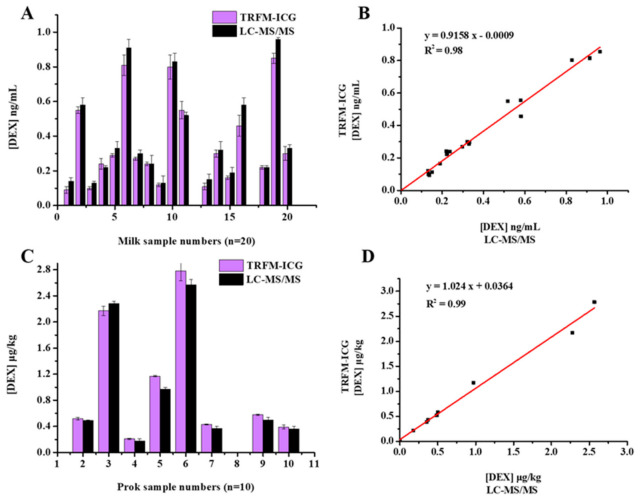
The correlation diagram of DEX detection results of the LC-MS/MS and TRFM-ICG: (**A**,**B**) in 20 milk samples; and (**C**,**D**) in 10 pork samples.

**Table 1 foods-10-01339-t001:** Recovery of the TRFM-ICG for the determination of DEX in milk and pork samples (*n* = 3).

Sample	Spiked Level (ng/mL or µg/kg)	Measured Level (ng/mL or µg/kg)	Recovery (%)	CV (%)
Milk	0.075	0.08 ± 0.01	106.7	12.5
	0.15	0.16 ± 0.01	106.6	6.3
	0.3	0.24 ± 0.03	80.0	12.5
Pork	0.35	0.28 ± 0.02	80.4	7.5
	0.7	0.55 ± 0.05	78.6	9.1
	1.4	1.17 ± 0.12	83.6	10.3

**Table 2 foods-10-01339-t002:** Determination of DEX in blind milk and pork samples by LC-MS/MS and TRFM-ICG (*n* = 3).

Sample	TRFM-ICG (ng/mL or µg/kg)	CV (%)	LC-MS/MS (ng/mL or µg/kg)	CV (%)
Milk 1	0.09 ± 0.01	11.1	0.14 ± 0.02	14.3
Milk 2	0.55 ± 0.02	3.6	0.58 ± 0.04	6.9
Milk 3	0.10 ± 0.01	10.0	0.13 ± 0.01	7.7
Milk 4	0.24 ± 0.03	12.5	0.22 ± 0.01	4.6
Milk 5	0.29 ± 0.01	3.5	0.33 ± 0.04	12.1
Milk 6	0.81 ± 0.06	7.4	0.91 ± 0.05	5.5
Milk 7	0.27 ± 0.01	3.7	0.30 ± 0.02	6.7
Milk 8	0.24 ± 0.01	4.2	0.24 ± 0.05	16.7
Milk 9	0.12 ± 0.01	8.3	0.13 ± 0.04	15.4
Milk 10	0.80 ± 0.07	8.8	0.83 ± 0.05	6.0
Milk 11	0.55 ± 0.05	9.1	0.52 ± 0.02	3.9
Milk 12	ND	-	ND	-
Milk 13	0.11 ± 0.02	18.2	0.15 ± 0.03	13.3
Milk 14	0.30 ± 0.04	13.3	0.32 ± 0.05	15.6
Milk 15	0.16 ± 0.02	12.5	0.19 ± 0.03	15.8
Milk 16	0.46 ± 0.06	13.0	0.58 ± 0.04	6.9
Milk 17	ND	-	ND	-
Milk 18	0.22 ± 0.01	4.6	0.22 ± 0.01	4.6
Milk 19	0.85 ± 0.03	3.5	0.96 ± 0.01	1.0
Milk 20	0.30 ± 0.04	13.3	0.33 ± 0.02	6.1
Pork 1	ND	-	ND	-
Pork 2	0.52 ± 0.02	3.9	0.49 ± 0.01	2.0
Pork 3	2.17 ± 0.07	3.2	2.28 ± 0.04	1.8
Pork 4	0.21 ± 0.01	4.8	0.18 ± 0.03	16.7
Pork 5	1.17 ± 0.01	0.9	0.97 ± 0.03	3.1
Pork 6	2.78 ± 0.15	5.4	2.57 ± 0.08	3.1
Pork 7	0.43 ± 0.01	2.3	0.37 ± 0.03	8.1
Pork 8	ND	-	ND	-
Pork 9	0.58 ± 0.01	1.7	0.50 ± 0.04	8.0
Pork 10	0.39 ± 0.03	7.7	0.36 ± 0.04	11.1

ND, not detected; -, unavailable.

**Table 3 foods-10-01339-t003:** Comparison of immunological methods for detecting DEX in animal-derived foods.

Method	Sample	Cut-Off Value (ng/mL or µg/kg)	LOQ (ng/mL or µg/kg)	Reference
CL-ELISA	Milk	-	0.3 (sample)	[36]
ELISA	Chicken muscle, liver	-	0.3, 0.5 (sample)	[38]
ELISA	Milk, liver	-	0.2, 0.6 (sample)	[37]
CG-ICG	Milk	0.5 (sample)	0.017 (buffer)	[25]
UCNP-ICG	Animal tissue	0.3 (sample)	0.05 (buffer)	[26]
LM-ICG	Milk, pork	0.3, 0.7 (sample)	0.047, 0.087 (sample)	[27]
TRFM-ICG	Milk, pork	0.25, 0.7 (sample)	0.003, 0.062 (sample)	This work

-, unavailable.

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
