# Peer review of "Portable, Rapid, and Sensitive Time-Resolved Fluorescence Immunochromatography for On-Site Detection of Dexamethasone in Milk and Pork"

_foods, 2021, doi:10.3390/foods10061339_

Round 1
Reviewer 1 Report
Li et al show a manuscript evaluating the quantitative detection of DEX in milk and pork using a time-resolved fluorescent microspheres immunochromatographic assay (TRFM-ICA) integrated with a portable fluorescence reader. It is a good work and the results support the conclusions but I have several minor comments (as shown in the attached pdf) related to grammatical errors as well as sentence structures.

Author Response
Response: We appreciate reviewer’s suggestion. We have revised the manuscript in accordance with the reviewer’s comments,the corrections in the revised manuscript have been marked in red.
Reviewer 2 Report
The manuscript is written properly and relatively easy to read.
My general comments are:
- there is a repetition in 2.2, are lines 108 and 113 providing different informations?
- Fig. 1C is not readable, this comment applied to all graphs. Axes description fonts are not sharp and to small.
To make a paper more attractive to general audience I suggest:
- to add description about TRMF fundamentals, e.g. how excitation and emission works and what are typical timescales of fluorescence one can observ.
- in paragraph 3.1.3 there are number of not extended acronyms which might make difficult to understand this text for non experts in the field.
Author Response
The manuscript is written properly and relatively easy to read.
My general comments are:
- there is a repetition in 2.2, are lines 108 and 113 providing different informations?
Response: We appreciate reviewer’s suggestion. There is not a repetition in 2.2. In the preparation of TRFM immunoprobe, two centrifugations are required. The first is after activation reaction, and the second is after blocking reaction. Lines 108 and 113 describe two different centrifugations.
- Fig. 1C is not readable, this comment applied to all graphs. Axes description fonts are not sharp and to small.
Response: We appreciate reviewer’s suggestion. Fig. 1C has been revised on page 3 of the revised manuscript, and the Axes description fonts are now clear.
To make a paper more attractive to general audience I suggest:
- to add description about TRMF fundamentals, e.g. how excitation and emission works and what are typical timescales of fluorescence one can observ.
Response: Thank you for your professional suggestion. The description about TRMF fundamentals has been added on line 70-73 of the revised manuscript.
- in paragraph 3.1.3 there are number of not extended acronyms which might make difficult to understand this text for non experts in the field.
Response: Thank you for your professional suggestion. The abbreviations in 3.1.3 have been annotated when they first appear in the article, but in order to make readers better understand, we add the abbreviation of the full text on page 12 of the revised manuscript.